# Epidemiology of Musculoskeletal Disorders among the Teaching Staff of the University of Douala, Cameroon: Association with Physical Activity Practice

**DOI:** 10.3390/ijerph18116004

**Published:** 2021-06-03

**Authors:** Archipe Mohamadou Tami, Elysée Claude Bika Lele, Jerson Mekoulou Ndongo, Clarisse Noel Ayina Ayina, Wiliam Richard Guessogo, Marie-Yvonne Lobe Tanga, Léon Jules Owona Manga, Abdou Temfemo, Bienvenu Bongue, Samuel Honoré Mandengue, Nathalie Barth, Peguy Brice Assomo Ndemba

**Affiliations:** 1Physiology and Medicine of Physical Activities and Sports Unit, University of Douala, Douala B.P. 2701, Cameroon; tamifosso@gmail.com (A.M.T.); claudebika@gmail.com (E.C.B.L.); meckjerson@yahoo.fr (J.M.N.); c_ayina@yahoo.fr (C.N.A.A.); guessowiliam@yahoo.fr (W.R.G.); mylt02@yahoo.fr (M.-Y.L.T.); owonaspinker@yahoo.fr (L.J.O.M.); temfemo@hotmail.com (A.T.); shmandengue@yahoo.fr (S.H.M.); 2National Institute of Youth and Sport, University of Yaounde I, Yaounde P.O. Box 1364, Cameroon; 3Department of Public Health, Faculty of Medicine and Pharmaceutical Sciences, University of Douala, Douala B.P. 2701, Cameroon; 4Laboratoire SAINBIOSE INSERM U1059, Dysfonction Vasculaire et Hémostase, Université Jean Monnet, 42023 Saint-Etienne, France; bienvenu.bongue@univ-st-etienne.fr (B.B.); nathalie.barth@univ-st-etienne.fr (N.B.); 5Department of Physiology, Faculty of Medicine and Biomedical Sciences, University of Yaounde I, Yaounde P.O. Box 1364, Cameroon

**Keywords:** musculoskeletal disorders, physical activity, university teachers

## Abstract

The aim of this study was to assess the epidemiology of musculoskeletal disorders (MSDs) among the teaching staff of the University of Douala and determine their association with physical activity (PA) practice. The Nordic questionnaire was used to assess MSDs. Ricci–Gagnon questionnaire was used to determine the level of PA. We recruited 104 participants mean-aged 42 ± 8 years, 80% male. Previous 7 days and 12 months prevalence were 56.7% and 80.8%, respectively. The most affected body regions were neck, shoulders and lower back. No significant association was found between MSDs and PA. Celibacy was significantly associated with previous 7-days MSDs (*p* = 0.048) while age ≥ 45 years and job seniority ≥ 10 years were significantly associated with a reduced risk of previous 12-months MSDs (*p* = 0.039 and *p* = 0.016, respectively). The prevalence of MSDs among university of Douala teaching staff showed no significant effect with the practice of PA.

## 1. Introduction

Musculoskeletal disorders (MSDs) are considered one of the most prevalent occupational health problems [1]. The World Health Organization (WHO) defines MSDs as any disorder affecting muscles, bones, joints, tendons and ligaments [2]. The musculoskeletal system is tough, but when exposed to infections, aging, heavy work load, repetitive stress, metabolic disorders and other factors, this causes tissue injuries [2,3,4]. These injuries lead initially to an inflammatory response then to pains and loss of motor function as an end result [4]. According to the WHO, in 2019 musculoskeletal conditions were the leading cause of disabilities worldwide and accounted for the greatest proportion of loss of productivity at the workplace [2].

The prevalence of MSDs worldwide is 20% to 33% [2]. It varies according to the type of activity [5,6,7,8,9,10] and affects both adolescents [11] and adults [5]. Teachers frequently present higher prevalence of MSDs, ranging from 39% to 95% [10,12]. Studies carried out among teaching staff in Africa showed 83.3% prevalence in secondary school teachers in Botswana [10]. Table 1 shows some studies carried on MSDs worldwide among teaching staff. Teachers are facing, during their work, multiple physical, intellectual and psychological constraints [10]. During the course of their work, teachers may be subjected to conditions that cause physical health problems such as long lasting upright standing position, head down posture and other awkward postures [10]. The teacher’s work does not only involve teaching students but also reading, preparing and marking of assignments, attending committees and being involved in extracurricular activities [13]. In addition, university teachers are often involved in research activities which are compulsory for their academic and professional progression.

In Cameroon, many situations can cause an increased burden of MSDs among university teaching staff, e.g., the increasing number of students, the necessity for some teachers to take out secondary job to make up with financial shortage, high work-related demand to meet up with new goals to compete with other universities. All these conditions involve a prolonged working in the standing position, computer work and carrying heavy loads that can lead to MSDs and limited physical activity practice.

Physical activity during work and leisure has been associated with reduction of cardiac diseases and other chronic conditions [23,24] but also with prevention of many occupational health problems including MSDs [25,26,27,28]. The increasing modernization and automation of the society has raised the occurrence of sedentary occupational tasks at work places, favoring the onset of all-causes morbidity and mortality [29]. Among university teaching staff situations, those that can favor sedentariness are: prolonged working time in the office or in the laboratories, increased number of tasks and activities (including teaching, doing administrative work, supervising student research activities) and other extracurricular activities. Previous studies have shown a relatively low level of physical activity among university teachers compared to other occupational sectors like secondary school teachers, medical and paramedical staff, private sector workers and army [28]. Although physical activity recommendations and strategies have been proposed to reduce occupational health problems [30], there is a lack of literature on the prevalence of MSDs amongst university teaching staff as well as its association with physical activity practice. The aim of this study was to determine the prevalence of MSDs amongst the teaching staff of the University of Douala and the impact of physical activity level on musculoskeletal complaints.

## 2. Materials and Methods

### 2.1. Study Design and Population

This was a cross-sectional analytical study conducted in the 11 Faculties and Higher Schools of the University for Douala (Littoral region, Cameroon). The University of Douala (the largest in Cameroon in terms of enrollment) counted 45,306 students registered during the 2016–2017 academic year for 771 permanent teaching staff members, giving a ratio of 58 students/01 teacher. All teaching staff members of all academic ranks with at least one year of experience were eligible to participate. Participants who refused to sign the informed consent and those with documented traumatic events in relation with MSDs were excluded. A minimum sample size of 92 participants was calculated using Lorentz formula with 96% prevalence of MSDs found in teachers [20], 95% confidence interval and 5% precision. The study was conducted in accordance with the Declaration of Helsinki and approved by the University of Douala Institutional and Ethical Board (Ethical clearance N° 2097 CEI-UDo/01/2020/T).

### 2.2. Data Collection

Data were collected using a structured questionnaire handed to participants.

The questionnaire was structured as: (i) socio-demographic characteristics (age, gender, marital status); (ii) socio-professional characteristics (grade, job seniority, number of hours of work/week) data; (iii) anthropometric parameters (height, weight, BMI, waist circumference); (iv) musculoskeletal disorders; and (v) physical activity practice.

### 2.3. Anthropometric Parameters

Height was measured using a fixed stadiometer and weight was measured using an electronic medical scale. Body mass index (BMI) was calculated as weight divided by the square of height. Participants were considered overweight when BMI ≥ 25 kg/m^2^ and obese when BMI≥ 30 kg/m^2^. Waist circumference (WC) was determined using a measuring tape and abdominal obesity was determined for WC ≥ 102 cm for men and WC ≥ 88 cm for women [31].

MSDs were determined using a modified version of the standardized Nordic Questionnaire [32]. The Nordic questionnaire was translated into French by one of the research team, then used to ensure the questionnaire functioned appropriately. Patients were asked if they have had complaints (pain, discomfort) during the previous 12 months and during the previous 7 days. For the localization of the complaints, participants had to choose from a photo showing body parts including the neck, shoulders, elbows, hands, upper back, lower back, hips, knees and feet.

The Ricci–Gagnon questionnaire [33] was used to assess the level of physical activity and sport practice. This questionnaire consists in two sub-sections A and B, each with four items. Each item is rated from 1 to 5: 1 corresponds to easy effort and 5, to a very difficult effort. Sub-section A evaluates the duration and intensity of daily common activities such as cleaning, gardening, rural work, walking. Sub-section B evaluates sports and recreational activities. The total of points in sub-sections A and B enable classification of participants as inactive if <16, active between 16 and 32, very active if >32.

### 2.4. Statistical Analysis

Data were computerized and analyzed using the SPSS 20 software (SSPS Inc, Chicago, Illinois, USA). Continuous variables are presented as mean and standard deviation and categorical data are presented as percentages. MSDs and physical activity were dichotomized as “present”/“absent” for MSDs and “active” (RG score ≥ 16)/“inactive” (RG score < 16) for physical activity practice, respectively, as previously defined [26]. The association of physical activity as well as other variables (as independent variables) with MSDs (as dependent variable) was assessed using multiple-adjusted logistic regression for any MSD and for MSD on each body region separately. Variables used in the multivariate model were age, gender, BMI and academic grade. Odds ratio (OR) and 95% confidence interval have been used to quantify the association between variables. Differences were set significant for *p* < 0.05.

## 3. Results

Of the 296 teachers who were asked to participate, 10 refused and of the 286 questionnaires handled, 110 were returned, making a response rate of 38%. The final sample for analysis included 104 participants who fulfilled all the requirements.

### Participants Characteristics

Table 2 shows socio-demographic, professional and anthropometric parameters of the participants. No participant was found physically very active considering the Ricci–Gagnon scale.

Figure 1 and Figure 2 show the prevalence of MSDs during the previous 7 days and the previous 12 months, respectively. Results are presented for all the participants as well for those with low and good physical activity levels. MSDs reported during the previous 7 days and during the previous 12 months represented 56.7% and 80.8%, respectively. Neck and low back were reported as the most affected body regions for MSDs during the previous 7 days (29.8% and 28.8%, respectively) and during the previous 12 months (53.1% and 48.1% respectively). Prevalence of MSDs in active vs. inactive participants during the previous 7 days and 12 months was 51.4% vs. 68.8% and 79.2% vs. 84.4%, respectively. However, the differences were not significant (*p* = 0.151 and *p* = 0.724, respectively). Likewise, differences were not significant between active and inactive participants for MSDs in different body regions during the previous 7 days and during the previous 12 months.

Table 3 shows multiple-adjusted association of MSDs with physical activity. Results are presented for MSDs reported during previous 7 days and during previous 12 months for all the body regions. No significant association was found between MSDs and physical activity for any body region affected in the 2 MSD variables before and after adjustment for cofounders.

Table 4 and Table 5 show association of MSDs with socio-demographic, socio-professional and anthropometric variables for MSDs reported during the previous 7 days and during the previous 12 months respectively. In univariate analysis, marital status was significantly associated with MSDs reported during the previous 7 days (Table 4), while age, marital status, academic grade and job seniority were significantly associated with MSDs reported during the previous 12 months (Table 5). After adjustment for confounders (age, gender, BMI and academic grade), marital status remained significantly associated with MSDs reported during the previous 7 days (OR: 2.31 95%CI 1.01–6.79, *p* = 0.048) while age higher or equal to 45 years (OR:0.15, 95%CI: 0.02–0.61, *p* = 0.039), marital status (OR 3.04; 95%CI 1.02–17.7; *p* = 0.021), job seniority higher or equal than 10 years (OR 0.13; 95%CI 0.01–0.72, *p* = 0.016) remained significantly associated with MSDs reported during the previous 12 months. The other variables were not significantly associated with MSDs.

## 4. Discussion

The aim of this study was to determine the prevalence of MSDs and for possible association between MSDs and physical activity among university teachers. We found a relatively high prevalence of MSDs in the population with 7-days prevalence and 12-months prevalence.

Contrary to what was hypothesized, we did not find any significant association of MSDs with physical activity despites the empirical differences observed between active and inactive participants in respective to MSDs prevalence. Seven-days and 12-months MSDs were independently associated with marital status for 7-days MSDs and with age, marital status and job seniority for 12-months MSDs. Although there is important literature on prevalence of MSDs, most of the studies focus only on MSDs reported either during the previous 12 months [19,22] or during the previous 7 days [8]. Our study was carried out on both previous 7-days and previous 12-months MSDs. This provides more reliable data on on-going process as well as past history of MSDs in the population. When looking at the relevance of previous 7-days and previous 12-months MSDs, our study shows that the prevalence of MSDs during the past 12 months was far higher than that of the previous 7 days. This is probably due to the fact that 12-months MSDs record all the evens during a longer period (1 year) than that of previous 7 days (1 week). The probability that one has experienced an MSD episode during 1 year is far higher than that of 1 week.

The comparison of our results with previous findings is difficult because of the scarcity of studies conducted on university teachers although numerous results found on primary and secondary school teachers. The prevalence of MSDs in this present study during the last 12 months was higher compared to previous studies made by Mohamed et al. among faculty members in Saudi Arabia (55%) [19], but lower that among of Brazilian university teachers (85.7%) [22] and Botswana secondary school teachers (83.3%) [10]. For the previous 7 days, the prevalence of MSDs in our study was higher compared to primary and secondary school teachers in Chuquisaca found in Bolivia (15%) [8]. University teaching is a highly physically demanding job [34], with an important work load (long hour, many subjects) [10]. Some studies report that physical factors such as prolonged standing, sitting, uncomfortable and awkward postures (i.e., when working on computer, writing on the board, reading books or theses) are known to be associated with increased prevalence of MSDs [6,10], and that may be carried out under unfavorable working conditions, especially in developing countries [10]. Most of these factors are currently observed in our population and can explain the higher prevalence of MSDs compared to other studies particularly in Saudi Arabia [19].

The most affected regions in our study were the neck, shoulders and low back which represented for 7-days MSDs 29.8%, 19.2% and 28.8%, respectively, and for 12-months MSDs 53.1%, 34.4% and 48.1%, respectively. Those three regions are commonly found as most affected in teachers by MSDs because of discomforts in the postures experienced during their activities [10,16]. High prevalence of MSDs in shoulders may be attributed to the fact that teachers commonly work at overhead positions with out-stretched arms, particularly writing on the board or pointing at images on boards/charts during teaching for a long time can cause friction, tension and strain over cervico-brachial regions. Similar are reported in previous studies [19,21,22].

For the last 12 months, the prevalence of MSDs in neck and shoulder regions were similar to what was found in university teachers in Saudi Arabia (53.3% and 23.6%, respectively) [19] and in Brazil (45.2% and 23.8%, respectively) [22] and in Pakistan (40% and 33.3%, respectively) [14]. Although our results were different from the above-mentioned studies, similar results have been found in Pakistan university teachers for the dorsal region, hips and thigh, knees, ankles and feet. The prevalence for MSDs in wrist and hand in our study (15.4%) was however lower compared to Saudi Arabia (31.6%) [19], Brazilian (23.8%) [35] and Pakistan University teachers (53.3%) [14]. University teachers in Douala are quite exposed to MSDs when compared to other above-mentioned university school’s results. For the last seven days the prevalence of MSDs in our study in the region of the neck was higher compared to University teachers in Pakistan (20%) [14] and lower than that of primary and secondary teachers in Bolivia (30.6%) [7] and Kenya (30.1%) [18]. Results on the shoulder and lumbar regions in our study were higher compared to results found in Bolivia [8], but lower compared to the university of Lahore [14] and Kenya (28.1%) [18]. The differences observed in prevalence in these different studies may be due to the study settings, population, sample size, workload and working conditions.

We found in our study a very low rate of physically active participants compared to the general population of workers. Actually, Mekoulou Ndongo et al. [28] found 63.6% physically active participants in a large sample of workers in Cameroon. This is more than twice higher than that we found in our study. Moreover, proportion of physically active participants was far higher in secondary education teachers (65.1%), private sector workers (59.1%) and paramedical staff (74.8%). These large differences compared to our study can be explained by the context of COVID-19 in which the study has been carried on. Actually, the recruitment of participants was carried during a period when Cameroon government has instituted a partial confinement, and this has probably influenced the responses of participants in relation to physical activity practice. In addition, the level of physical activity recorded in the study of Mekoulou Ndongo et al. [28] among university teachers was still lower than that of the other working sectors. This sedentariness has been previously associated with burnout syndrome in that population [27] and underlines the necessity to encourage university teachers for more implication in physical activity programs that are known to reduce stress and prevent occupational health problems [25].

Contrary to what we expected, we did not find any significant association between physical activity and MSDs both on 7-days and 12-months MSDs. However, we observed a higher prevalence of 7-days MSDs in participants who were inactive compared to those who were active. This was particularly remarkable for any MSD presence (68.8% vs. 51.4%) as well as for neck (40.6% vs. 25%), upper back (18.8% vs. 12.5%) and lower back (34.4% vs. 24.4%). A comparable study had already found a significantly protective effect of physical activity on burnout syndrome among 303 teaching staff members of the University of Douala [27]. Likewise, physical activity was also significantly associated with a lower prevalence of dorsal pain in secondary teachers in Botswana [10] and with any MSDs in female teachers in Saudi Arabia [1]. The possible reasons that can explain these results is that exercises can improve strength, flexibility, pain threshold and makes muscles and ligaments stronger for optimal functioning and prevents injury [17]. This is particularly useful for teachers who frequently use awkward and long-lasting standing positions. Although a study reported a protective effect of physical activity on MSDs [36], other studies found that sport practice have a negative impact on locomotor system because of the frequent injuries observed during sport practice; particularly among participants with low physical fitness or those who are not involved in a professional physical activity. In addition to being more likely to suffer injuries, exercising too much can leave one feeling weak and tired [37,38]. The questionnaire we used in this study evaluated home and leisure physical activities as well as sport practice. Further studies are necessary to determine the relative impact of different circumstances of physical activity practice on MSDs. The relatively low sample size of our study can also explain the non-significance of our results.

Association of MSDs with other variables of the study shows a significant increased risk of MSDs in unmarried compared to married participants both for 7-days and 12-months MSDs (*p* = 0.048 and *p* = 0.021, respectively). For the 12-months MSDs, we also found a significantly reduced risk of MSDs with age, particularly in participants ages 45 years or older compared to those less than 35 years (*p* = 0.039) as well as a reduced risk in participants with more or equal to 10 years job seniority compared to those with one to five years (*p* = 0.016). These results corroborate the very-low prevalence of 12-months MSDs observed in Professors (60%) compared to Lecturers (80.4%) and Senior Lecturers (88.4%) although the difference was not significant in multivariable analysis. The prevalence of MSDs in Professors in our study was lower compared to study on Brazilian Professors (85.7%) [35]. Other studies did not mention the prevalence of MSDs according to grade. The significant drop of prevalence of MSDs with age as well as with job seniority can be due to the fact that older teachers usually have more responsibilities which demand less physical efforts than younger ones. A couple of study showed that younger workers face greater work demands, being exposed to risk factors, as they take over more activities and tasks in the beginning of the career [15,20]. Other researchers found similar results to our study [1,8,10,17]. These results appeal the necessity of public health policies to focus on MSDs as well as other occupational health problems in younger workers and those at the starting of their career.

### Limits

There are some limitations identified in this study. As the study was cross-sectional, we cannot draw conclusion on the causal relationship between physical activity and MSDs. The evaluation of physical activity was done using a questionnaire which is known to have a limited validity and reliability. However, this method is commonly used and is practical on working participants. The Ricci–Gagnon questionnaire has been used because it has already been used in some studies [27,39,40,41] and it is known to be very comprehensive and easy to fill. Moreover, in the situation of partial confinement due to the COVID-19, the International Physical Activity Questionnaire (IPAQ) was not appropriate to record the habitual physical activity practice of participants because it focuses only on the physical activity done during the previous 7 days.

## 5. Conclusions

The findings of this study suggest that the prevalence of previous 7-days and 12-months MSDs among university teaching staff were independently associated with marital status, age and job seniority. We did not find significant association of MSDs with physical activity both for previous 7-days and 12-months MSDs. Some useful preventive measures should be applied at the level of the university to decrease the high prevalence of musculoskeletal pain, such as proportional reduction of workload and increasing the number of teachers.

## Figures and Tables

**Figure 1 ijerph-18-06004-f001:**
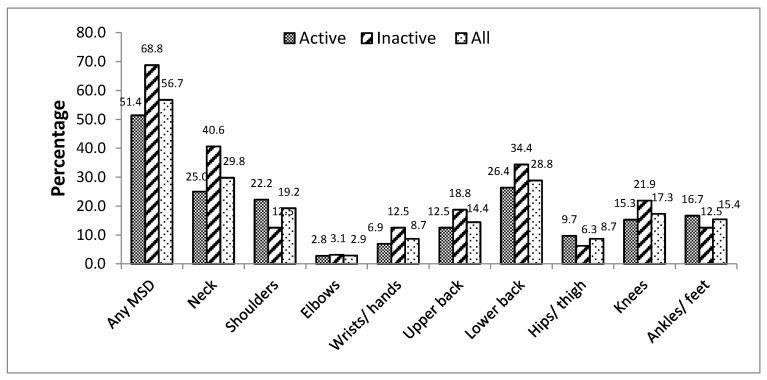
Distribution of MSDs during previous 7 days according to physical activity.

**Figure 2 ijerph-18-06004-f002:**
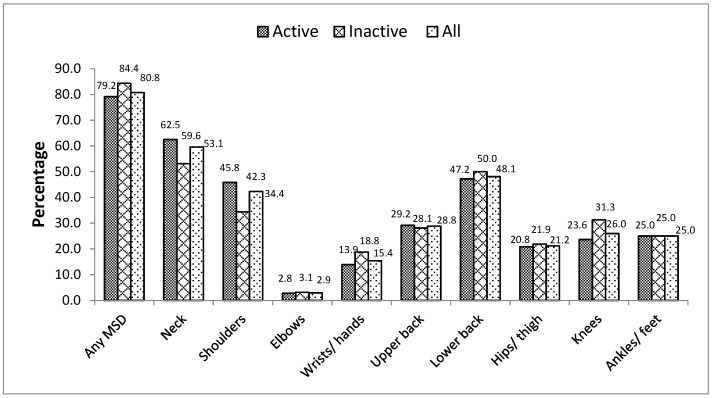
Distribution of MSDs during previous 12 months according to physical activity.

**Table 1 ijerph-18-06004-t001:** Musculoskeletal disorders among teachers in various countries.

Authors	Number of Participants	Country	Study Population	Relevant Findings
Mahmood [14]	150	Pakistan	University teachers	Previous 12-months prevalence: wrist and hands (53.3%), lumbar region (66.7%), hip/thigh (53.3%), (73.3%).
Elias et al. [15]	417	Kenya	Primary school	The previous 12-months prevalence: low back (64.9%). 70% minimal disability.
			teachers	
Temesgen et al. [16]	754	Ethiopia	Primary and secondary school teachers	Previous 12-months prevalence: shoulder and/ neck (57.3%)
Abdel-Salam et al. [1]	254	Saudi Arabia	Secondary school	Previous 12-months prevalence: Any site (68.5%); low back (68.4%); knee (58.6%); shoulder (47.7%); neck (45.4%); elbow (23.6%); wrist (14.4%). Days of absenteeism per month among (89.7%). Associated factors: age > 40 years, not practicing exercise, more than 10 years of teaching and non-comfortableness of school furniture.
			Female teachers	
Ojukwu et al. [17]	352	Nigeria	primary and secondary school teachers	Work-related MSDs: Any site (70.2%); shoulder (62.3%); neck (57.9%).
Ndonye et al. [18]	302	Kenya	Primary school teachers	Work-related MSDs: Any site (85.1%); lower back (58.6%); knees (57.6%); neck (53.3%); ankles (53%). Associated factor: age, teaching for over four hours while standing, teaching for over four hours while sitting, working on a head-down posture and lack of back support on chairs.
Sirajudeen et al. [19]	60	Saudi Arabia	University teachers	Previous 12-months MSDs: Any site (55%); neck (53.5%); lower back (43.3%) hands and wrists (31.6%). Associated factors: Computer use and lack of ergonomic training.
El Gendy et al. [20]	200	Egypt	Preparatory school	Previous 12-months prevalence: Any site (96%); neck and back (83.5%).
			teachers	
Solis-Soto et al. [8]	1062	Bolivia	preschool, primary	Previous 12-months prevalence: Any site (86%); neck (47%); wrist/hands (26%). Previous 7-days prevalence: overall (63%). Associated factors: rural (vs. urban) school.
			secondary and university teachers	
Modarresi et al. [21]	113	Iran	University Teachers	Previous 12-months prevalence: Any site (84.1%); waist (27.9%); neck (25%); wrist/fingers (16.3%); work-related disability (23%).
Junior and Silva [22]	49	Brazil	University teachers	Previous 12-months prevalence: Any site (85.7%); lumbar spine (54.8%); cervical spine (45.2%); shoulders (23.8%); wrists/hands (23.8%).
Erick et al. [10]	1732	Botswana	Primary and secondary school teachers	Previous 12-months prevalence: Any site (83.3%); dorsal region (52.6%); shoulder (52.5%); neck (50.8%); ankle/feet (37.8%); hips/thighs (18.2%).

**Table 2 ijerph-18-06004-t002:** Characteristics of study participants.

		Continuous Mean ± SD	Categorical N (%)
Age (years)	Mean ± SD	42 ± 8	
Marital status	Single		27 (27.3)
	Married		72 (72.7)
Height (m)	Mean ± SD	1.73 ± 0.08	
Weight (kg)	Mean ± SD	80 ± 14.5	
BMI (kg/m^2^)	Mean ± SD	27 ± 3.9	
Overweight/obesity	No		31 (29.4)
	Yes		73 (70.6)
Waist circumference (cm)	Mean ± SD	92 ± 11	
Abdominal obesity	No		71 (68.3)
	Yes		33 (31.7)
Grade	Assistant		43 (41.3)
	Lecturer		46 (44.2)
	Professor		15 (14.4)
Job Seniority (years)	Mean ± SD	10 ± 7	
	1–5		31 (29.8)
	5–10		33 (31.7)
	≥10		40 (38.5)
Number of working time h/week	Mean ± SD	14 ± 12	
	<20		78 (75)
	≥20		26 (25)
Physical activity	Inactive		72 (69.2)
	Active		32 (30.8)

**Table 3 ijerph-18-06004-t003:** Association of MSDs during the previous 7 days and the previous 12 months with physical activity practice.

	MSDs during Previous 7 Days		MSDs during Previous 12 Months	
	Crude OR ^a^ (95%CI)	Adjusted OR ^a,b^ (95%CI)	Crude OR ^a^ (95% CI)	Adjusted OR ^a,b^ (95% CI)
Any MSD	0.48 (0.20–1.16)	0.51 (0.20–1.27)	0.70 (0.23–2.14)	1.09 (0.33–3.64)
Neck	0.49 (0.20–1.18)	0.52 (0.20–1.34)	1.47 (0.63–3.41)	2.00 (0.80–5.02)
Shoulders	2.00 (0.61–6.55)	2.28 (0.67–7.80)	1.62 (0.68–3.83)	2.06 (0.81–5.22)
Elbows	0.89 (0.08–10.14)	0.94 (0.04–23.67)	0.70 (0.23–2.12)	0.94 (0.04–23.67)
Wrists/hands	0.52 (0.13–2.09)	0.36 (0.07–1.80)	1.05 (0.42–2.65)	0.71 (0.21–2.40)
Upper back	0.62 (0.20–1.92)	0.81 (0.24–2.70)	1.05 (0.42–2.65)	1.28 (0.48–3.42)
Lower back	0.68 (0.28–1.68)	0.76 (0.28–2.02)	0.89 (0.39–2.06)	0.93 (0.38–2.30)
Hips/thigh	1.62 (0.32–8.24)	1.68 (0.30–9.34)	0.94 (0.34–2.59)	1.04 (0.32–3.36)
Knees	0.64 (0.22–1.85)	0.68 (0.23–2.05)	0.68 (0.27–1.71)	0.67 (0.25–1.76)
Ankles/feet	1.40 (0.41–4.73)	1.67 (0.46–6.06)	1.00 (0.38–2.62)	1.13 (0.41–3.12)

^a^: Reference category: inactive (OR = 1); ^b^: Adjusted by age, gender, BMI and academic grade; OR: odds ratio; CI: confidence interval.

**Table 4 ijerph-18-06004-t004:** Association of MSDs during previous 7 days with socio-demographic, socio-professional and anthropometric variables.

		N (%)	Crude OR (95%CI)	*p*	Adjusted OR (95% CI)	*p*
Age (years)	< 35	15 (71.4)		0.228		0.231
	35–44	32 (56.1)	0.51 (0.17–1.51)	0.225	0.47 (0.14–1.54)	0.212
	≥ 45	12 (46.2)	0.34 (0.10–1.16)	0.086	0.28 (0.06–1.21)	0.087
Gender	Female	15 (71.4)				
	Male	44 (53.0)	0.44 (0.16–1.25)	0.123	2.30 (0.77–6.85)	0.134
Marital status	Married	36 (50.0)				
	Single	23 (71.9)	2.56 (1.04–6.28)	0.041	2.31 (1.01–6.79)	0.048
Grade	Assistant	26 (60.5)		0.810		0.742
	Lecturer	25 (54.3)	0.78 (0.34–1.81)	0.560	1.35 (0.50–3.63)	0.552
	Professor	8 (53.3)	0.75 (0.23–2.44)	0.630	1.70 (0.41–7.04)	0.467
Job seniority (years)	1–5	22 (71.0)		0.160		0.210
	5–10	16 (48.5)	0.39 (0.14–1.08)	0.070	0.33 (0.08–1.31)	0.115
	≥10	21 (52.5)	0.45 (0.17–1.22)	0.117	0.64 (0.14–2.98)	0.568
Teaching time h/week	<20	45 (57.7)				
	≥20	14 (53.8)	0.86 (0.35–2.09)	0.732	0.82 (0.32–2.12)	0.689
Overweight/obesity	No	20 (65.0)				
	Yes	36 (50.0)	0.54 (0.18–1.58)	0.261	0.56 (0.18–1.72)	0.311
Abdominal obesity	No	42 (57.1)				
	Yes	19 (57.7)	1.02 (0.40–2.62)	0.963	1.32 (0.43–4.05)	0.629

OR: odd ratio; CI: confidence interval.

**Table 5 ijerph-18-06004-t005:** Association of MSDs during previous 12 months with socio-demographic, socio-professional and anthropometric variables.

		N (%)	Crude OR (95% CI)	*p*	Adjusted OR (95% CI)	*p*
Age (years)	<35	19 (90.5)		0.005		0.032
	35–44	50 (87.7)	0.75 (0.14–3.95)	0.736	0.73 (0.12–4.57)	0.733
	≥45	15 (57.7)	0.14 (0.03–0.75)	0.021	0.15 (0.02–0. 61)	0.039
Gender	Female	19 (90.5)				
	Male	65 (78.3)	2.63 (0.56–12.4)	0.221	2.35 (0.44–12.7)	0.320
Marital status	Married	54 (75.0)				
	Single	30 (93.8)	5.00 (1.09–23.0)	0.039	3.04 (1.02–17.7)	0.021
Grade	Assistant	38 (88.4)		0.072		0.681
	Lecturer	37 (80.4)	0.54 (0.17–1.77)	0.309	1.05 (0.25–4.36)	0.950
	Professor	9 (60.0)	0.20 (0.05–0.79)	0.022	0.57 (0.10–3.14)	0.519
Job seniority (y)	1–5	30 (96.8)		0.077		0.169
	5–10	25 (75.8)	0.10 (0.01–0.89)	0.039	0.09 (0.01–1.19)	0.068
	≥10	29 (72.5)	0.09 (0.01–0.72)	0.024	0.13 (0.01–0.72)	0.016
Teaching time h/week	<20	65 (83.3)				
	≥20	19 (73.1)	0.54 (0.19–1.55)	0.255	0.64 (0.17–2.36)	0.500
Overweight/obesity	No	23 (75.0)				
	Yes	58 (79.2)	1.27 (0.37–4.33)	0.706	1.58 (0.41–6.11)	0.509
Abdominal obesity	No	60 (83.9)				
	Yes	24 (73.1)	0.52 (0.17–1.60)	0.253	0.51 (0.12–2.14)	0.359

OR: odds ratio; CI: confidence interval.

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
