# Peer review of "Epidemiology of Musculoskeletal Disorders among the Teaching Staff of the University of Douala, Cameroon: Association with Physical Activity Practice"

_ijerph, 2021, doi:10.3390/ijerph18116004_

Round 1

Reviewer 1 Report

This is very interesting work.  The data adds to the knowledge base of the prevalence of MSDs in university teaching staff. 

The following items, however, should be addressed before publication (most are minor):

  • Identify the study as retrospective
  • When referring to 7 days and 12 months the terminology “previous 7 days and previous 12 months” provide clarity. The terminology “previous 7 days and previous 12 months” is used in some places in the manuscript but not in others, and should be included in the abstract
  • In the methods section it states that the Ricci-Gangon survey can provide three categories of exercise, but only two categories of exercise are presented in the results
  • In the results section the median number of hours worked per week is given as 10 hours and the mean number of hours worked per week is given as 14 + In Table 3, 14 individuals (53.8%) report working more that 20 hours per week during the previous 7 days while 45 individuals (57.7%) report working less than 20 hours per week during the previous 7 days.   In table 4, 19 individuals (73.1%) of individuals report working more than 20 hours per week during the previous 12 months and 65 (83.3%) of individuals report working less than 20 hours per week during the previous 12 months.  The hours worked in the Results section don’t match the numbers for hours worked in Tables 3 and 4.  The numbers for hours worked in tables 3 and 4 are not internally consistent.
  • More specific description on statistical analysis needs to be included. For example
    • Without the data on the number of teachers who did not have any MSDs who exercised and the number of teachers who did not have MSDs who did not exercise the reader cannot calculate the odds ratio
    • More information on how the confidence interval was calculated
    • What test was used to calculate p values
  • Great information on other studies on MSDs in teachers in various countries. It might be easier to understand if it were put in a table so that the results from the various countries could be easily compared with your results. 
  • Why do you think there is such a difference between the number of MSDs reported for the previous 7 days compared with the number reported for the previous 12 months? Do teachers receive treatment for their MSDs or do they get better without any medical intervention?
  • Does the Ricce-Gangon survey distinguish between aerobic and non-aerobic exercise? A little more discussion about the exercise survey would be helpful.

Line 75                  Change “on” to “one”

Line 91                  insert a comma after the parentheses, remove “and” and insert “activity” after professional

Line 92                  Insert “and” before musculoskeletal

Line 111                change “to classify” to “classification of”

Author Response

Please see the attchment

Regards

Peguy Brice ASSOMO NDEMBA (corresponging author)

Reviewer 2 Report

General

The authors examine an interesting issue and one that is worth of study. However, there are a number of issues which need development in order to develop the manuscript further. The authors need to make the argument for their study more robustly in the introduction and then analyse it more robustly in the discussion. Why do we really need to know the MSD prevalence in just over 100 university teachers from Cameroon? Is there something specific about the Cameroonian teaching system that means it might be different than Brazil, or Saudi, or any of the other countries the authors cite prior studies from?

Likewise, really what can we learn from this size sample. The authors need to get into this in more depth and make a convincing argument here. Simply stating sample size as a limitation is not really sufficient. Then suggesting staff were ‘too busy’ to complete the questionnaire also is not convincing, what was done to go back and re request staff t complete, what methods were used to encourage completion, etc. This is not to discourage the authors but more to signal more detail, and analysis is needed.

I also suggest some specific points below that need consideration

Specific

Abstract: please place musculoskeletal disorders in full on first use

Introduction, line 37: what do the authors mean by ‘This system’ the musculoskeletal system? If so, would be worth stating explicitly

Line 48-49: sentence mentioning constraints, grammatically this could do with a restructure to aid flow better and as the authors make a direct statement here it would be worth including a reference to support these claims.

In this first paragraph of the introduction, the focus is on teachers and only at the end do university teachers feature. I can see this is part of argument development, but it needs to be made clear that teachers and university teachers are different things, especially in regard to demand son the musculoskeletal system

Line 57: delete ‘physical activity’ after  ‘leisure time’

In the last paragraph in the argument for examining university teachers needs to be made more strongly. For example there is work showing high levels of sedentary behaviour https://www.tandfonline.com/doi/abs/10.1080/10803548.2021.1874704?tab=permissions&scroll=top which could lead to MSK problems? Perhaps the authors might look at this and other similar studies to construct their argument for the need for such monitoring.

Line 101: can the authors explain how the Nordic questionnaire was modified?

Line 106: are there any reliability/validity data available for the Ricci-Gagnon questionnaire? If so would be good to state.

Results, first line: this could be rewritten to better express the process. Of the 296 teachers who were asked to participants, 10 refused, resulting in a total sample of xxxxx

Overall, I think a discussion needs to be made in regard to whether the reporting of MSDs in the preceding 7 days is a valid reflection of overall ‘load’ on the musculoskeletal system, or is this just a snapshot?

Line 212: would be useful to give an example in terms of the postures experienced during their activities. Eg through excessive computer work? Or another type of activity?

Line 233: ‘Contrary to what expected, we did not found any’ change found to find

Line 252: ‘was evaluating’ change to ‘evaluated’

In this section of the discussion, the authors should get into the data around physical activity and sedentary behaviour a little more. Overall are we talking about a population who are generally more sedentary? If so then the less active teachers would be very sedentary? Or is the sample population very active overall? Again, if so the inactive group would be relatively active.  In relative terms how does this sample compare to the general public overall? From there you can make inference about how active/inactive each of the groups are in your study, which may then explain more about the prevalence of MSDs.

Line 278: if the authors knew the questionnaire had limited reliability/validity, it begs the question why was it used? Including a justification in this section of why this questionnaire (over other more reliable self reports) was used is important. For example, surely the IPAQ might have provided a more robust estimate of PA?

Line 280-281: repetition of ‘limitation’ and ‘limited’ in the same sentence

Line 284 im assuming ‘4.discussion’ is an error

Line 285-288: im not sure this should be here, I suspect its from the MDPI template and needs removing.

Conclusion, line 290: relatively high to what? Please identify how you ascertained the rate was relatively high, compared to what? National norms?

Line 291: ‘find’ not ‘found’

Line 293: ‘were’ not ‘was’

Author Response

Regards

Peguy Brice ASSOMO NDEMBA (corresponding author)

Round 2

Reviewer 2 Report

The authors have tried hard to address comments on the previous draft and I appreciate them doing so. However, there were a number of points which were not fully addressed, were only addressed in the response to reviewer (and not the manuscript), or perhaps the point has been missed.

In the methods be clear that the Nordic questionnaire was translated into French. If this was translated by one of the research team, please be clear on that and be clear if back translation was then used to ensure the  questionnaire functioned appropriately.

Line 243: ‘explain by the context of COVID-19’  should be ‘explained’

Line 248: ‘sedentariness have been’  ‘has’ not ‘have’

In the limitations section the IPAQ is mentioned, which is fine but this was only an example, there are many other questionnaires. Some of the responses to the original review mention that the Ricci-Gagnon questionnaire has ‘been used in many studies in that population’ if this is the case please cite them as justification. From the authors’ response it seems there is no validity or reliability data available for this measure. Therefore the authors need to be clear that this is the case in the limitations.

Conclusions: I am afraid my original comment still stands. The authors suggest prevalence  of MSDs is high but make no attempt to identify how they know they are ‘high’, high compared to what? Other studies, national statistics on MSDs? The authors need to provide some way to determine how ‘high’ prevalence is. There is also a need to answer the ‘so what’ question in the conclusion. I am not suggesting the topic is not worthy of examination but what is the key outcome of the research. Essentially, the researchers suggest university teachers are at increased risk of MSDs because of their job requirements related to being sedentary, sitting, standing and so on. What about this area did we not know previously? It is not sufficient to state the it’s the first study from Cameroon on this topic as, while interesting, whats so different about university teaching in Cameroon compared to say the UK, US, Uganda, Ghana, Belgium, etc, etc, that makes it important we look at this topic in this population. This needs addressing in order to make the manuscript more applicable outside of Cameroon.

Line 305: ‘finds’ not ‘find’
